**Data Availability Statement:** All relevant data are within the manuscript and its Supporting Information files.

# The impact of community based continuous training project on improving couples' knowledge on birth preparedness and complication readiness in rural setting Tanzania; A controlled quasi-experimental study

**Fabiola V. Moshi** [1]*, **Stephen M. Kibusi**[2], **Flora Masumbuo Fabian**[3]

**1** Department of Nursing and Midwifery, College of Health Sciences of University of Dodoma, Dodoma, Tanzania, **2** Department of Public Health, College of Health Sciences of University of Dodoma, Dodoma, Tanzania, **3** Department of Biomedical Sciences, College of Health Sciences of University of Dodoma, Dodoma, Tanzania

* fabiola.moshi@gmail.com

## Abstract

### Background

It is widely accepted that community-based interventions are vital strategies towards reduction of maternal and neonatal mortalities in developing counties. This study aimed at finding the impact a Community Based Continuous Training (CBCT) project in improving couples' knowledge on birth preparedness and complication readiness in rural Tanzania.

### Method

The quasi-experimental study design with control was adopted to determine the impact of CBCT in improving knowledge on birth preparedness and complication readiness. The study was conducted from June 2017 until March 2018. A multi-stage sampling technique was employed to obtain 561couples. Pre-test and post-training intervention information were collected using semi-structured questionnaires. The impact of CBCT was determined using both independent t-test and paired t-test. Linear regression analysis was used to establish the association between the project and the change in knowledge mean scores. The effect size was calculated using Cohen's d.

### Results

At post-test assessment, knowledge mean scores were significantly higher in the intervention group among both pregnant women (m = 14.47±5.49) and their male partners (m = 14.1 ±5.76) as compared to control group among both pregnant women (m = 9.09±6.44) and their male partners (m = 9.98±6.65) with large effect size of 0.9 among pregnant women and medium effect size of 0.66 among male respondents. When the mean scores were compared within groups among both pregnant women and male partners in the intervention

**Funding:** This work is part of my PhD project. The University of Dodoma sponsored the tuition fee and field allowance. Publication fee was not covered in the sponsorship. The funders had no role in decision to publish, or preparation of the manuscript The authors received no specific funding for this work.

**Competing interests:** No authors have competing interests

group, there were a significant increase in knowledge mean scores at post-test assessment as compared to pre-test assessment with large effect size of Cohen's d = 1.4 among pregnant women and 1.5 among male partners. After adjusting for the confounders, the predictors of change in knowledge among pregnant women were the CBCT project ($\beta$ = 0.346, p<0.000) and ethnic group [Mambwe ($\beta$ = -0.524, p = 0.001)] and the predictors of change in knowledge among male partners were the CBCT project ($\beta$ = 1.058, p<0.001) and walking distance [more than five kilometers ($\beta$ = -0.55, p< 0.05)].

## Conclusion

This interventional study which focused on knowledge empowerment and behavior change among expecting couples was both feasible and effective on improving knowledge about birth preparedness and complication readiness in rural settings of Tanzania.

## Introduction

It is estimated that 830 women died every day due to complications of pregnancy and childbirth in 2015 worldwide [1]. The trend of maternal mortality has shown a decrease from 385 maternal deaths per 100,000 live births in 1990 to 216 maternal deaths per 100,000 live births in 2015 [2]. The majority of these deaths occur in developing regions which accounted for approximately 99% (302 000) of the global maternal deaths in 2015, with sub-Saharan Africa alone accounting for roughly 66% (201 000), followed by Southern Asia 30% (66 000) [1]. Tanzania is among the countries in sub Saharan Africa with highest maternal mortalities which is estimated to 556 maternal deaths per 100,000 live [2], meaning that for every 1,000 live births in Tanzania, almost 6 women die due to pregnancy related causes. There exists a regional variation within Tanzania with Rukwa region leading with the highest maternal mortality of 860 maternal deaths per 100,000 live births [3].

The sustainable development goal target 3.1 has targeted to reduce the global maternal mortality ratio (MMR) from 216 per 100 000 live births in 2015 to less than 70 per 100 000 live births by 2030. To achieve this target, it requires a global annual rate of reduction of at least 7.5%–which is more than three times the annual rate of reduction that was achieved between 1990 and 2015 [4].

Studies have reported that there is a direct relationship between maternal services utilization and birth outcomes [5]. The use of maternal services in sub Saharan Africa including Tanzania where maternal and neonatal mortalities are high is unacceptably low [6, 7]. Birth preparedness and complication readiness(BPCR) have a potential to improve maternal services utilization in low resources countries [8]. BPCR is when a mother or couple engages in advance planning and preparation for childbirth, including unanticipated emergencies. Its components include preparation for normal delivery, readiness to deal with complications and postnatal and newborn care preparations [9]. The components of BPCR include planning for place of childbirth and identification of a skilled birth attendant, make arrangement for transport to be used in case emergency, mobilize resources to assist during childbirth, identify a relative who will donate a blood in case of emergency, knowing when to book for antenatal clinic and the recommended number of visits and knowing the danger signs during pregnancy, Labor and childbirth, during postnatal period and neonatal danger signs. It has been reported that birth preparedness can significantly reduce the three delays to access maternal services [10]. The delays are; the first delay(delay in decision making), second delay(delay to reach

health facility) and third delay(delay to obtain appropriate care upon reaching health facility) [10]. Despite the reported potentials of birth preparedness on reducing the three delays, the practice of birth preparedness in developing countries including Tanzania is still low [11–13]. For example, a study conducted in Tanzania reported low birth preparedness among expecting mothers as only 0.8% identified skilled birth attendants, 10.2% identified transport and 47.2% saved money for emergency [11]. Low levels of birth preparedness were also reported on similar studies conducted in Nigeria and Uganda [12, 13].

Knowledge about birth preparedness and complication readiness is a key factor in achieving BPCR. Studies have revealed that there are low knowledge on BPCR among both pregnant women and their male partners [14]. Studies have also revealed that training interventions which include male partners can improve pregnant women's knowledge about BPCR [10].

It has been also revealed that, when male partners are empowered with necessary information about birth preparedness and complication readiness the preparation improves [15]. This is largely because most of families in developing countries male partners control the family earnings. When they are empowered on obstetric danger signs and what to prepare in order to facilitate timely access to emergency or delivery services, there will be a significant improvement of BPCR. For many years, health education and information about the progress of pregnancy and childbirth in low resources settings like Tanzania were given to women and men left aside. This strengthens the perception that pregnancy and childbirth are responsibility of women, men's responsibility is to provide financial support upon required [16].

There have been several interventions which seek to address the issue of maternal and neonatal mortalities. Some interventions like Information and Education for Empowerment and Change (IEEC) have focused on improving birth preparedness and birth outcomes. It was a community-based intervention geared towards the reduction of maternal and neonatal mortalities. The project trained women and their husbands on the identification of obstetric danger signs. The intervention was effective on increasing the utilization of health facilities for prenatal services and during obstetric complications [17].

Similarly, a community-based intervention focusing on training community on birth preparedness showed a significant increase on knowledge, attitude and practice of birth preparedness in Eritrea. The rate of antenatal attendance also showed a significant increase compared to the comparison group. The proportion of women who delivered in a health facility increased from 3% to 47% in the intervention group compared to an increase of 4% to 15% in the comparison group [18].

Also, another community-based intervention was done in rural Tanzania to evaluate the effectiveness of home-based life saving program on improving knowledge of danger signs during pregnancy, childbirth and postpartum, preparedness for childbirth and use of health facilities for childbirth. The project reported a significant increase in knowledge, birth preparedness and increased use of health facility for childbirth [19].

The community-based continuous training project went further and used the theory of planned behavior, the involvement of intimate partners and knowledge empowerment to improve birth preparedness and complication readiness among expecting couples in rural settings of Rukwa Tanzania. Theory of Planned Behavior (TPB) explores the relationship between behavior and beliefs, attitudes, and intentions. The TPB is a theoretical model of behavior change where a central factor in the theory is the individual's intention to perform a given behavior [19]. It implies that the intention to perform a certain behavior is influenced by three domains, attitude, perceived subjective norms and perceived behavior control. Attitudes, subjective norms and perceived behavior control are assumed to be based on a corresponding set of beliefs [20]. Individuals will have the intention to perform a behavior when they evaluate it positively, believe that others think they should perform it, and perceive it to be within their

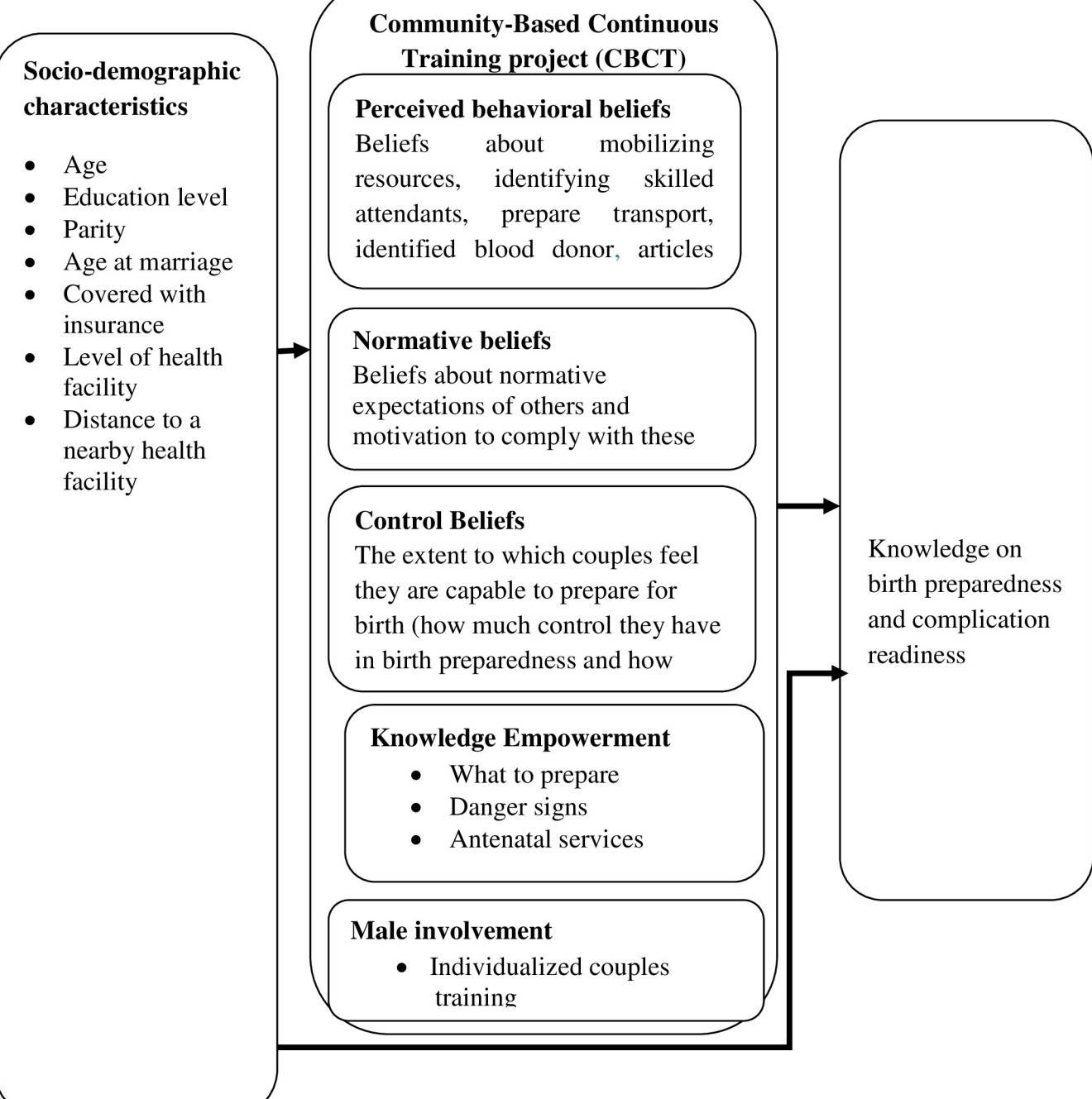

**Fig 1. Community-based continuous training (CBCT).**

control [20]. Intention to engage in birth preparedness itself is not sufficient in the occurrence of the behavior. Also, the project added knowledge empowerment component which covered what to prepare, danger signs and antenatal services and male involvement in behavior change programs and knowledge empowerment (Fig 1).

Most couples ignore early warning signs due to a lack of adequate knowledge and information about danger signs during pregnancy and labor and so delay to seek care [21]. Men in

developing countries have little knowledge regarding reproductive health compared to their female partners but have much decision-making power which ultimately affects timely access to obstetric care [22]. Moreover, men in these countries decide the timing and conditions of sexual relations, family size, and whether or not their spouses utilize available health care services, all of which have a significant impact on maternal and neonatal outcomes.

Little was known about the effectiveness of the CBCT project on improving knowledge about birth preparedness and complication readiness among pregnant women and their male partners in rural Tanzania.

## Method

This section of the method has been partly published previously in the protocol titled The Effectiveness of Community-Based Continuous Training on Promoting Positive Behaviors towards Birth Preparedness, Male Involvement, and Maternal Services Utilization among Expecting Couples in Rukwa, Tanzania: A Theory of Planned Behavior Quasi-Experimental Study [23].

### Study design

A quasi-experimental study with control using pre-post-tests was conducted to evaluate the effect of the CBCT project on improving knowledge on BPCR in a rural setting of Rukwa Tanzania from June 2017 to March 2018 [23].

### Setting

The quasi-experimental study was conducted among expecting couples in Rukwa Region in the Southern Highlands of Tanzania. The region had a population of 1,004,539 people; 487,311 males and 517,228 females. The forecast for 2014 was 1,076,087 persons with a growth rate of 3.5%. The region has the lowest mean age at a marriage where males marry at the age of 23.3 years and 19.9 years for females and a fertility rate of 7.3 [24].

### Sample size and sampling method

**Sample size calculation.** The sample size for couples who were involved in the study was calculated using the following formula [25]

$$n = \frac{\left\{Z\alpha\sqrt{[\pi o(1-\pi o)]} + 2\beta\sqrt{[\pi 1(1-\pi 1)]}\right\}^2}{(\pi 1 - \pi o)^2}$$

Where:
n = minimum sample size
Zα = Standard normal deviation (1.96) at 95% confidence level for this study
2β = standard normal deviate (0.84) with a power of demonstrating a statistically significant difference before and after the intervention between the two groups at 90%
πo = Proportion at pre- intervention (Use of skilled delivery in Rukwa region 65%) [2]
π1 = proportion after intervention (Proportion of families which will access skilled birth attendant 75%)

$$n = \frac{\left\{1.96\sqrt{[0.65(1-0.65]} + 0.84\sqrt{[0.75(1-0.75)]}\right\}^2}{(0.75-0.65)^2}$$

n = 169 couples + 10% = 187

Therefore, the required sample size in the intervention group = 187 couples

Intervention: control ratio = 1:2 (using age groups in five years and parity) which aimed at increasing comparability of these two groups. Therefore, the sample size in the control group = 374 couples.

## Sampling technique

Two districts (Sumbawanga Rural District and Kalambo District) were purposively selected from the four districts of the Rukwa Region using the criterion of high home childbirth. Three stages of sampling technique (multistage) were used to enroll study participants in the study. In the first stage, all wards (12 wards of Sumbawanga Rural District and 17 wards of Kalambo District) were listed and a simple random sampling using the lottery method was used to obtain five wards from Sumbawanga rural district and ten wards from Kalambo rural district. In the second stage sampling, all villages from selected wards from each district were listed and a simple random sampling was used to obtain fifteen villages from Sumbawanga Rural District and thirty villages from Kalambo Rural District. The systematic sampling technique was finally used to obtain households that met the inclusion criteria (pregnancy of 24 weeks or below and living with a male partner). In each visited household, a female partner was assessed for the signs and symptoms of pregnancy. Research assistants who were nurses assessed for pregnancy. A female partner who had missed her period for two months was requested to complete a pregnancy test. Those with positive tests who gave consent to participate were enrolled in the study. If the visited household did not have eligible participants, the household was skipped and researchers entered into the next household.

## The CBCT project procedure

Step one of the project was pre training assessment of expecting couples. In step two, a total of fifteen village health workers were recruited for two days of training to empower them on knowledge about birth preparedness and complication readiness. On the second day, the village health workers discussed the available misconceptions about danger signs and how to address them during couple training. The training was concluded by providing training manual and reading materials for trainees (expecting couples). In step three, the actual couples' training occurred where each expecting couple was visited for two days. Towards the end of the second day, they were assessed for comprehension of the sessions. The fourth step of the project was immediate assessment of knowledge after training. The fifth step was about the signs of labor and neonatal care. The teaching was initiated by revising the previous learned subject-birth preparedness then concluded by a discussion of misconceptions about signs of labor. The actual intervention was empowering expecting couples on danger signs and what to prepare for birth and obstetric emergencies. Teaching and learning materials were prepared using pictures to enhance understanding. Trainers were prepared on how to deliver the sessions and were provided with facilitation guides. Learners' guides were prepared and were given to them for references. It was individualized couple's training where each couple were trained at their households and were assisted to initiate birth plan (Fig 2).

## Data collection procedure

Semi-structured questionnaires were used for data collection. Trained research assistants interviewed the study participants and recorded their responses. Research assistants collected data both at pre-test and post-test assessment. Trained community health workers facilitated the four intervention sessions. The primary investigator coordinated the whole process of data collection. The first part had questions to measure knowledge of birth preparedness and

**Step One**

**Pre-training assessment** of expecting Couples June and July 2017

**Step Two**
Prepare community health workers to be trainers of couples by:
-**Empowering community** health workers on knowledge about birth preparedness
-**Discuss the local beliefs** which hinder birth preparedness, male involvement and maternal services utilization
-**Empower ho**w to develop a birth plan

**Step Three**
**First Session ≤ 24 GA**
Provide individualized couples' training by Empower couples on:
-Knowledge on birth preparedness
-Development of birth plan
-Correct misconceptions (belief barriers)

**Step Five**

**Monitor**

o  Antenatal attendance
o  Screened for HIV and Syphilis
o  Male involvement
o  Items for birth prepared
o  Transport prepared
o  Saved money
o  Timely arrive to health facility during labor
o  Health facility childbirth
o  Mother and child checkups after seven days post delivery
o  Child immunization 42 days post delivery

**Step Five**
**Second Session GA ≥28 GA**
To empower couples on
o  signs of labor
o  Neonatal care (breast feeding, care of umbilical cord, cleanliness)
o  Diet during postnatal

**Step four**
Post-training assessment **October 2017 Knowledge on BPCR immediate assessment**

**Step Six**
**Final Assessment (February and March 2018)**
-  **Knowledge on BPCR**
-  **BPCR**
-  **Male involvements**
-  **Maternal services Utilization**

**Fig 2. The CBCT project procedure.**

complication readiness, questions were adapted and modified from monitoring BPCR tools for maternal and newborn health [26]. Several studies have adopted this tool [27, 28]. Pre-test assessment in both groups was done in June and July 2017, the intervention group was visited in October 2017 for immediate assessment of the intervention which was done from July to September. The post-test assessment in each group was done in February and March 2018.

### Data processing and analysis

The collected data were assessed for completeness and consistencies, then they were coded and entered in to computer using statistical package IBM SPSS version 23. Frequency distribution and cross-tabulation were obtained using descriptive statistical analyses were used to describe the characteristic of the study participants. The linear logistic regression model was used to determine the predictors of change in knowledge mean scores from the pre-training assessment and post-training assessment. Both paired and independent t-test was used to measure the differences in mean scores between the pre-training and post-training assessment. The effect size of the differences in mean scores was determined using a Cohen's d test.

### Ethical consideration

The proposal was approved by the Ethical Review Committee of the University of Dodoma. A letter of permission was obtained from the Rukwa Regional Administration. Both written and verbal consent was sought from study participants after explaining the study objectives and procedures. A couple was included into the study if both of them consented to participate in the study. Their right to refuse to participate in the study at any time was assured.

## Results

Majority of pregnant women in both the intervention (n = 110, 58.8%) and the control group (n = 224, 59.9%) were younger than 25 years, while majority of male respondents in both intervention (n = 123, 65.8%) and the control group (n = 261, 69.8%) were older than 25 years. Majority of pregnant women in both intervention (n = 134, 71.7%) and control n = 270, 72.1% were married at an age of less than 18 years, while (n = 116, 61.7% in the intervention and n = 208, 55.6% in the control group) including male partners (n = 135, 72.2% in the intervention and n = 269, 71.9% in the control group), had at least a primary education (Table 1). Most of them earned less than one dollar per day (64% in the intervention and 76.7% in the control group), whom received a basic obstetric care services from dispensaries (75.4% in the intervention and 86.9% in the control), were not covered by health insurance while (78.1% in the intervention as well 61.8% in the control), lived less than five kilometers a walking distance to a nearby health facility (93.5% in the intervention and 88% in the control group) Table 1.

### Obstetric characteristics of pregnant women

The majority of pregnant women (78.6% in the intervention group and 78.1% in the control group) were multiparous. Most of the pregnant women had neither a history of preterm delivery (94.1% in the intervention and 94.9% in the control) nor the history of caesarian section (98.4% in the intervention and 97.1% in the control) Table 2.

### Knowledge about birth preparedness and complication readiness at pretest assessment

On recalling obstetric danger signs among pregnant women; 50.8% in the intervention group and 51.9% in the control group were not able to recall any danger sign during pregnancy.

**Table 1. Distribution of pregnant women and their male partners by socio-demographic characteristics (N = 1122).**

| Variable | Pregnant Women | | Male Partners | |
|---|---|---|---|---|
| | Intervention n = 187, n(%) | Control n = 374 n(%) | Intervention n = 187, n(%) | Control n = 374 n(%) |
| Age (years) | | | | |
| Less than 20 | 55 (29.4) | 119 (31.8) | 14 (7.5) | 14 (3.7) |
| 20 to 25 | 55 (29.4) | 105 (28.1) | 50 (26.7) | 99 (26.5) |
| 26 to 30 | 39 (20.9) | 67 (17.9) | 40 (21.4) | 110 (29.4) |
| 31 to 35 | 15 (8.02) | 40 (10.7) | 40 (21.4) | 49 (13.1) |
| 36 and above | 23 (12.3) | 43 (11.5) | 43 (23) | 102 (27.3) |
| Age at Marriage (years) | | | | |
| Less than 18 | 134 (71.7) | 270 (72.2) | 27 (14.4) | 47 (12.6) |
| 19 to 24 | 50 (26.7) | 103 (27.5) | 114(61) | 250 (66.8) |
| 25 and above | 3 (1.6) | 1(0.3) | 46 (24.6) | 77 (20.6) |
| Marital status | | | | |
| Cohabit | 46 (24.6) | 111(29.7) | 46 (24.6) | 111(29.7) |
| Married | 141 (75.4) | 263 (70.3) | 141(75.4) | 263 (70.3) |
| Education level | | | | |
| Non-formal | 71 (38) | 166 (44.4) | 52 (27.8) | 105 (28.1) |
| Primary School | 111 (59.4) | 196 (52.4) | 121(64.7) | 244 (65.2) |
| Secondary school or Higher | 5 (2.3) | 12 (3.2) | 14 (7.5) | 25(6.7) |
| Ethnic group | | | | |
| Fipa | 155 (82.9) | 178 (47.6) | 175 (93.6) | 203 (54) |
| Mambwe | 5 (2.3) | 119 (31.8) | 3 (1.6) | 117 (31.3) |
| Others | 27 (14.4) | 77 (20.6) | 9 (4.8) | 54 (14.4) |
| Income per day | | | | |
| Less than 1 dollar | 120(64.2) | 287 (76.7) | 122 (65.2) | 271 (72.5) |
| At least one | 67(35.8) | 87 (23.3) | 65 (34.8) | 103 (27.5) |
| Health Insurance | | | | |
| Yes | 41(21.9) | 143(38.2) | 41 (21.9) | 133 (35.6) |
| No | 146(78.1) | 231(61.8) | 146 (78.1) | 241 (64.4) |
| Health facility | | | | |
| Dispensary | 141(75.4) | 325(86.9) | 141(75.4) | 325 (86.9) |
| Health centre | 46 (24.6) | 49(13.1) | 46 (24.6) | 49 (13.1) |
| Distance to health facility | | | | |
| Less than 1 | 74 (39.6) | 188 (50.3) | 74 (39.6) | 188 (50.3) |
| 1 to 5 | 102 (54.5) | 141(37.7) | 102 (54.5) | 141(37.7) |
| More than 5 | 11(5.9) | 45 (12) | 11 (5.9) | 45 (12) |
| Own mobile phone | | | | |
| Yes | 16 (8.6) | 53 (14.2) | 91(48.7) | 149(39.8) |
| No | 171 (91.4) | 321(85.8) | 96 (51.3) | 225(60.2) |
| Own a radio | | | | |
| Yes | 95 (50.8) | 218 (58.3) | 95 (50.8) | 218(58.3) |
| No | 92 (49.2) | 156 (41.7) | 92 (49.2) | 156(41.7) |

Also, 62% of pregnant women in the intervention and 64.7% in the control group were not able to recall any danger signs during labor and childbirth, 72.2% in the intervention and 67.9% in the control were not able to recall any danger sign during 42days post-delivery, 67.4% in the intervention and 62.9% in the control during neonatal period at pre-test assessment. Majority of male partners in the intervention 67.4%, 71.7%, 82.9% and 73.3% failed to

**Table 2. Obstetric history of the female respondents (N = 561).**

| Variable | Intervention Group (n1, %) n = 187 | Control group (n2, %) n = 374 | Total (n1+n2) n = 561 |
|---|---|---|---|
| *Gestation Age (Weeks)* | | | |
| 8–16 | 29(15.5) | 90(24.1) | 119(21.2) |
| 17–24 | 158(84.5) | 284(75.9) | 442(78.8) |
| *Gravidity* | | | |
| Prime-gravid | 40(21.4) | 82(21.9) | 122(21.75) |
| Multiparous | 147(78.6) | 292(78.1) | 439(78.3) |
| *Parity* | | | |
| Null-parous | 40(21.4) | 82(21.9) | 122(21.75) |
| Para 1–4 | 109(58.3) | 221(59.1) | 330(58.82) |
| Para 5+ | 38(20.3) | 71(18.9) | 109(19.43) |
| *History of pre-term delivery* | | | |
| Yes | 11(5.9) | 19(10.2) | 30(5.35) |
| No | 176(94.1) | 355(94.9) | 531(94.65) |
| *History of Caesarean section* | | | |
| Yes | 3(1.6) | 11(2.9) | 14(2.49) |
| No | 184(98.4) | 363(97.1) | 547(97.5) |

recall any danger sign during pregnancy, labor and childbirth, postnatal period and neonatal period respectively. In the control group 46%, 53.2%, 65.5% and 58% of the male respondents, failed to recall any danger sign during pregnancy, labor and childbirth, postnatal period, and to neonates respectively. On recalling the preparations to be made for childbirth, only 2.1% in the intervention and 7.2% in the control, managed to mention more than three preparations (Table 3).

When knowledge mean scores were compared between groups, only knowledge on components of birth preparedness showed a significant increase in knowledge means score among pregnant women in the control group as compared to pregnant women in the intervention group at pre-test assessment. At post-test, all variables of birth preparedness showed a significant mean score increase in the intervention group as compared to the control group (Table 4; Fig 3).

At posttest, pregnant women in the intervention group had significantly higher mean scores (M = 14.47±5.489) as compared to pregnant women in the control group (M = 9.07 ±6.438); p<0.001 with a large effect size of Cohen's d = 0.9.

However, at the pre-test male respondents from the control group had significantly higher knowledge mean scores (M = 5.5±5.207), as compared to male respondents in the intervention group (4.43±2.748); p = 0.002 with a small effect size of Cohen's = 0.2 (Table 5).

Furthermore, at the posttest, there were significant differences in the mean knowledge scores in the intervention group (M = 14.1±5.761) and control group (M = 9.98±6.654); t-test = 7.48 at 95% CI = 3.03–5.207, 9<0.001, with a medium effect size of Cohen's d = 0.66 (Table 5; Fig 4).

## Comparison within groups

When the mean scores were compared within groups among pregnant women in the intervention group; the results showed a significant increase in knowledge mean scores at post-test assessment (M = 14.47±5.487) as compared to pre-test assessment (M = 4.53±4.416); p<0.001 with large effect size Cohen's d = 1.4 (Table 6).

In the control group, there was a significant increase in knowledge mean score at post-test assessment (M = 9.07±6.438) as compared to pre-test assessment (M = 4.94±4.383); p<0.001 with a medium effect size of Cohen's d = 0.5 (Table 6).

**Table 3. Knowledge about birth preparedness and complication readiness at pre-test assessment.**

| Variable | Pregnant Women | | | Male Partners | | |
|---|---|---|---|---|---|---|
| | Intervention | Control | P-Value | Intervention | Control | P-Value |
| | n(%) | n(%) | | n(%) | n(%) | |
| Danger signs **pregnancy** | | | | | | |
| Unable to recall any danger sign | 95(50.8) | 196(52.4) | | 126(67.4) | 172(46) | |
| Mentioned 1 to 3 | 83(44.4) | 163(43.6) | | 57(30.5) | 182(48.7) | |
| Mentioned more than 3 | 9(4.8) | 15(4) | 0.869 | 4(2.1) | 20(5.3) | 0.000 |
| **Danger signs labor and childbirth** | | | | | | |
| Unable to recall any danger sign | 116(62) | 242(64.7) | | 134(71.7) | 199(53.2) | |
| Mentioned 1 to 3 | 68(36.4) | 127(33.95) | | 53(28.3) | 172(46) | |
| Mentioned more than 3 | 3(1.6) | 5(1.3) | 0.781 | 0(0) | 3(0.8) | 0.000 |
| **Danger signs during 42 days post-delivery** | | | | | | |
| Unable to recall any danger sign | 135(72.2) | 254(67.9) | | 155(82.9) | 245(65.5) | |
| Mentioned 1 to 3 | 49(26.2) | 107(28.6) | | 32(17.1) | 109(29.1) | |
| Mentioned more than 3 | 3(1.6) | 16(4.3) | 0.218 | 0(0.0) | 20(5.3) | 0.000 |
| **Danger signs during the neonatal period** | | | | | | |
| Unable to recall any danger sign | 126(67.4) | 227(60.7) | | 137(73.3) | 217(58) | |
| Mentioned 1 to 3 | 57(30.5) | 139(37.2) | | 50(26.7) | 146(39) | |
| Mentioned more than 3 | 4(2.1) | 8(2.1) | 0.223 | 0(0.0) | 11(3) | 0.001 |
| **Components of birth preparedness** | | | | | | |
| Unable to recall any component | 86(46) | 109(29.1) | | 28(15) | 150(40.1) | |
| Recalled 1 to 3 signs | 96(51.3) | 247(66) | | 155(82.9) | 197(52.7) | |
| Recalled more than 3 signs | 5(2.7) | 18(4.8) | 0.001 | 4(2.1) | 27(7.2) | 0.000 |
| **When (weeks) a pregnant woman has to start antenatal clinic?** | | | | | | |
| Incorrect | 86(47.3) | 155(42.6) | | 45(24.1) | 185(49.5) | |
| Correct | 96(52.7) | 209(57.4) | 0.315 | 142(75.9) | 189(50.5) | 0.000 |
| **What are the recommended numbers of antenatal visits?** | | | | | | |
| Incorrect | 69(37.9) | 119(32.7) | | 45(24.1) | 168(44.9) | |
| Correct | 113(62.1) | 245(67.3) | 0.227 | 142(75.9) | 206(55.1) | 0.000 |

As for the male respondents in the intervention group, there was a significant increase in the knowledge mean scores at the post-test (M = 14.1±5.761), as compared to the knowledge mean scores at the pre-test assessment (M = 4.43±2.748); p<0.001, with a large effect size of Cohen's d = 1.5 (Table 7).

In the control group there was also a significant increase in the knowledge mean scores at the post-test assessment (M = 9.98±6.654), as compared to the pre-test assessment (M = 5.5 ±5.207); p<0.001 with a medium size effect of Cohen's d = 0.57 (Table 7).

## Predictors of change in knowledge score between pre-test and post-test assessment among pregnant women and their male partners

Simple linear regression was calculated to establish the variables which influenced change before adjusting for confounders. The variables which showed a significant influence among pregnant women were being in the intervention group (p<0.001), ethnic group [Mambwe, (p<0.05)], age at marriage (p<0.01) and having ever heard about birth preparedness (p<0.05) Table 8.

Variables among male respondents which showed significant influence with the change in the knowledge score were the intervention group (p<0.001), age of respondent (p<0.05),

**Table 4. Effectiveness of CBCT project on improving knowledge (comparison between groups) among pregnant women.**

| Variable | Intervention Mean (SD) | Control Mean (SD) | t-test | 95% CI | | p-value |
|---|---|---|---|---|---|---|
| | | | | Lower | Upper | |
| *Pre-test assessment* | | | | | | |
| *Danger signs* | | | | | | |
| Pregnancy | 0.89(1.13) | 0.88(1.14) | 0.134 | -0.188 | 0.216 | 0.894 |
| Labor and childbirth | 0.59(0.91) | 0.54(0.88) | 0.638 | -0.109 | 0.213 | 0.524 |
| Post-childbirth | 0.43(0.86) | 0.57(1.07) | -1.626 | -0.303 | 0.029 | 0.105 |
| Neonatal period | 0.65(1.11) | 0.69(1.05) | -0.361 | -0.23 | 0.159 | 0.719 |
| Components of BP | 0.82(1.01) | 1.02(1.05) | -2.132 | -0.38 | -0.015 | 0.034 |
| Antenatal services | 1.14(0.84) | 1.25(0.81) | -1.459 | -0.258 | 0.038 | 0145 |
| Knowledge on BPCR | 4.53(4.42) | 4.94(4.38) | -1.037 | -1.201 | 0.372 | 0.300 |
| *Post-test assessment* | | | | | | |
| *Danger signs* | | | | | | |
| Pregnancy | 3.2(1.26) | 2.09(1.57) | 8.941 | 0.866 | 1.354 | 0.000 |
| Labor and childbirth | 2.2(1.24) | 1.01(1.16) | 10.735 | 0.967 | 1.401 | 0.000 |
| Post-childbirth | 2.18(1.35) | 1.4(1.32) | 6.378 | 0.538 | 1.017 | 0.000 |
| Neonatal period | 2.55(1.39) | 1.76(1.41) | 6.301 | 0.55 | 1.049 | 0.000 |
| Components of BP | 2.94(1.48) | 1.85(1.57) | 7.951 | 0.821 | 1.36 | 0.000 |
| Antenatal services | 1.39(0.83) | 0.96(0.92) | 5.551 | 0.28 | 0.588 | 0.000 |
| Knowledge on BPCR | 14.47(5.49) | 9.07(6.44) | 10.21 | 4.357 | 6.434 | 0.000 |

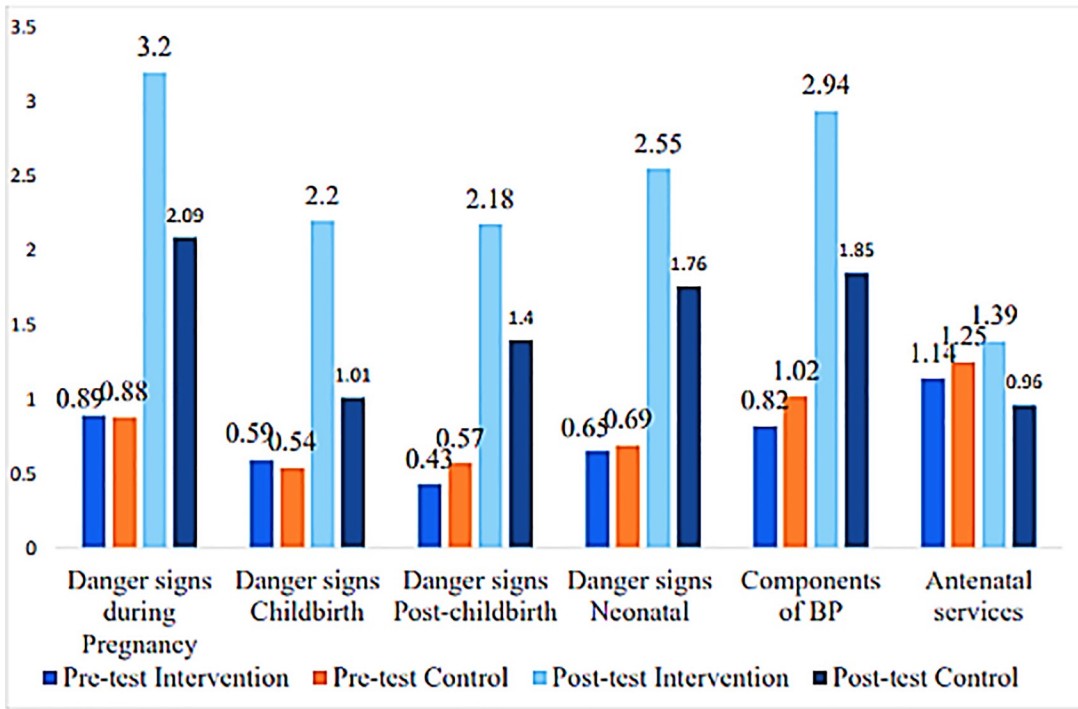

**Fig 3. Knowledge mean scores compared between groups among pregnant women.**

**Table 5. Effectiveness of CBCT project on improving knowledge (comparison between groups) among male respondents.**

| Variable | Intervention | Control | Mean | t-test | 95%CI | | p-value |
|---|---|---|---|---|---|---|---|
| | Mean (SD) | Mean (SD) | diff. | | Lower | Upper | |
| **Pre-test assessment** | | | | | | | |
| Danger signs during pregnancy | 0.62(1.064) | 1.13(1.310) | -0.514 | -4.912 | -0.719 | -0.308 | 0.000 |
| During labor | 0.4(0.727) | 0.81(1.052) | -0.409 | -5.308 | -0.561 | -0.258 | 0.000 |
| 42 days post-childbirth | 0.24(0.599) | 0.68(1.179) | -0.448 | -5.886 | -0.597 | -0.298 | 0.000 |
| Neonatal period | 0.38(0.732) | 0.82(1.152) | -0.434 | -5.345 | -0.594 | -0.275 | 0.000 |
| Components of birth preparedness | 1.03(0.685) | 1(1.189) | 0.03 | 0.376 | -0.128 | 0.188 | 0.707 |
| Antenatal services | 1.77(0.587) | 1.06(0.83) | 0.712 | 11.57 | 0.591 | 0.832 | 0.000 |
| Knowledge on BPCR | 4.43(2.748) | 5.5(5.207) | -1.063 | -3.122 | -1.732 | -0.394 | 0.002 |
| **Endline assessment** | | | | | | | |
| Danger signs during pregnancy | 3.02(1.272) | 2.24(1.567) | 0.78 | 6.238 | 0.534 | 1.026 | 0.000 |
| During labor | 2.12(1.306) | 1.27(1.233) | 0.843 | 7.247 | 0.614 | 1.072 | 0.000 |
| 42 days post-childbirth | 2.11(1.378) | 1.5(1.368) | 0.613 | 4.909 | 0.367 | 0.858 | 0.000 |
| Neonatal period | 2.42(1.461) | 1.95(1.492) | 0.475 | 3.558 | 0.213 | 0.738 | 0.000 |
| Components of birth preparedness | 2.88(1.415) | 1.95(1.575) | 0.931 | 6.977 | 0.669 | 1.194 | 0.000 |
| Antenatal services | 1.55(0.74) | 1.07(0.92) | 0.481 | 6.584 | 0.337 | 0.624 | 0.000 |
| Knowledge on BPCR | 14.1(5.761) | 9.98(6.654) | 4.124 | 7.48 | 3.04 | 5.207 | 0.000 |

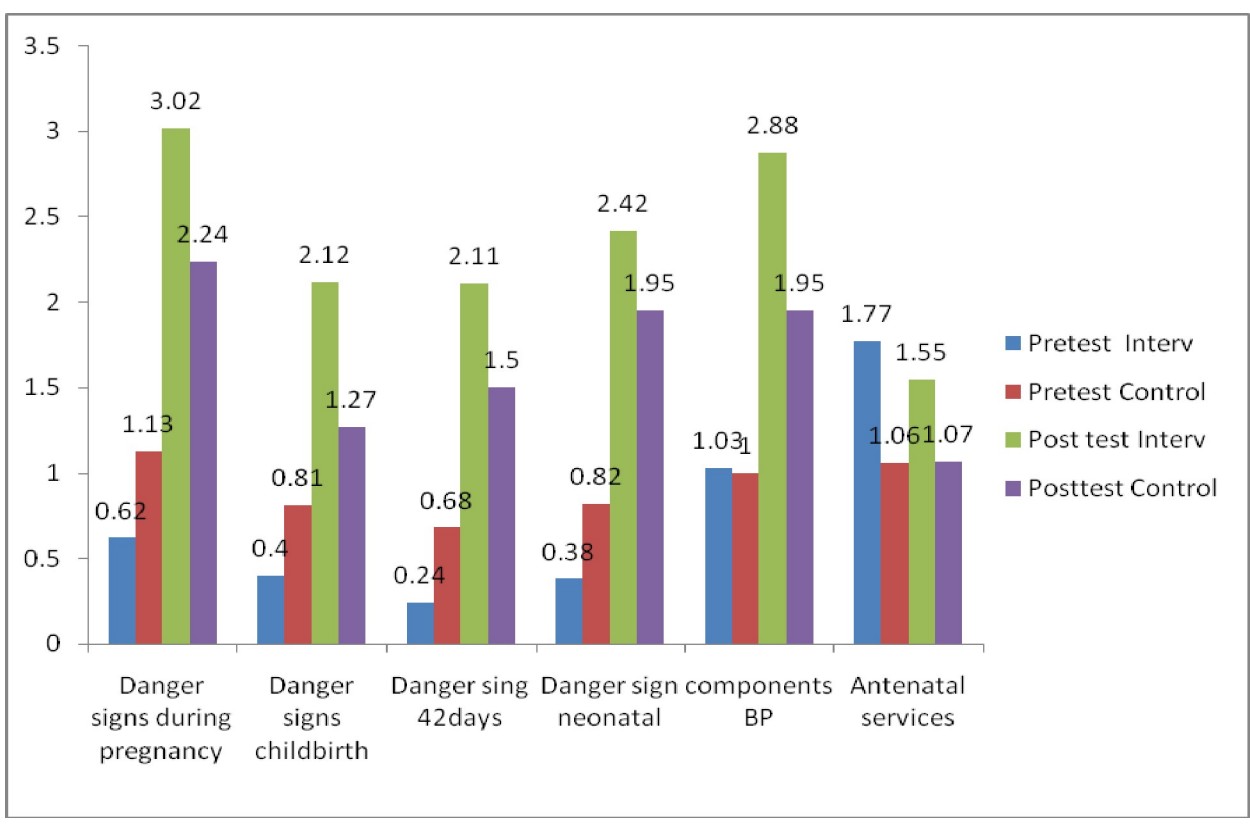

**Fig 4. Knowledge mean scores compared between groups among male partners.**

**Table 6. Effectiveness of CBCT project on improving knowledge (comparison within groups) among pregnant women.**

| | Post-test | Pre-test | t-test | 95% CI | | p-value |
|---|---|---|---|---|---|---|
| Variable | Mean (SD) | Mean (SD) | | Lower | Upper | |
| **Knowledge on Danger signs during** | | | | | | |
| Pregnancy-Intervention | 3.02 (1.272) | 0.62 (1.064) | 17.82 | 2.057 | 2.569 | 0.000 |
| Pregnancy-Control | 2.09 (1.567) | 0.88 (1.14) | 12.05 | 1.018 | 1.416 | 0.000 |
| Labor -Intervention | 2.2 (1.241) | 0.59 (0.91) | 14.72 | 1.389 | 1.82 | 0.000 |
| Labor -Control | 1.01(1.161) | 0.54(0.882) | 6.303 | 0.325 | 0.62 | 0.000 |
| 42 days -Intervention | 2.18 (1.352) | 0.43(0.856) | 14.52 | 1.515 | 1.991 | 0.000 |
| 42days post-Control | 1.4 (1.323) | 0.57(1.065) | 9.366 | 0.662 | 1.014 | 0.000 |
| Neonatal -Intervention | 2.55 (1.392) | 0.65(1.11) | 14.4 | 1.641 | 2.162 | 0.000 |
| Neonatal -Control | 1.76 (1.408) | 0.69(1.05) | 11.25 | 0.88 | 1.252 | 0.000 |
| **Knowledge on Components** | | | | | | |
| Components -Intervention | 2.94 (1.48) | 0.82(1.009) | 15.54 | 1.847 | 2.384 | 0.000 |
| Components -Control | 1.85 (1.572) | 1.02(1.047) | 8.706 | 0.64 | 1.014 | 0.000 |
| **Knowledge on antenatal** | | | | | | |
| Antenatal -Intervention | 1.39 (0.832) | 1.14(0.84) | 2.935 | 0.083 | 0.423 | 0.004 |
| Antenatal -Control | 0.96 (0.917) | 1.25(0.809) | -4.74 | -0.41 | -0.17 | 0.000 |
| **Total Knowledge on BPCR** | | | | | | |
| Knowledge on BPCR-Intervention | 14.47 (5.487) | 4.53(4.416) | 18.86 | 8.899 | 10.98 | 0.000 |
| Knowledge on BPCR-Control | 9.07 (6.438) | 4.94(4.383) | 10.32 | 3.342 | 4.916 | 0.000 |

ethnic group [Mambwe,(p<0.001)], characteristic of the nearby health facility [dispensary (p<0.05)], walking distance to the health facility [one to five kilometers (p<0.05), more than five kilometers (p<0.05)] Table 9.

In the multiple linear regression, the predictors of change in the knowledge scores among pregnant women were the intervention (β = 0.346, p<0.000) and the ethnic group [Mambwe (β = -0.524, p = 0.001)]. Among male respondents, predictors of change were the intervention

**Table 7. The effectiveness of CBCT project on improving knowledge (comparison within groups) among male respondents.**

| Variable | Post-test | Pre-test | t-test | 95% CI | | p-value |
|---|---|---|---|---|---|---|
| | Mean (SD) | Mean (SD) | | Lower | Upper | |
| **Knowledge on danger signs** | | | | | | |
| pregnancy-Intervention | 3.02 (1.272) | 0.62 (1.064) | 19.69 | 2.161 | 2.642 | 0.000 |
| pregnancy-Control | 2.24 (1.567) | 1.13 (1.31) | 11.5 | 0.918 | 1.296 | 0.000 |
| labor and childbirth-Intervention | 2.12 (1.306) | 0.4 (0.727) | 15.61 | 1.498 | 1.931 | 0.000 |
| labor and childbirth-Control | 1.27 (1.233) | 0.81 (1.052) | 5.459 | 0.295 | 0.628 | 0.000 |
| 42days post-childbirth-Intervention | 2.11 (1.378) | 0.24 (0.599) | 17.09 | 1.657 | 2.09 | 0.000 |
| 42days post-childbirth-Control | 1.5 (1.368) | 0.68 (1.179) | 9.205 | 0.639 | 0.987 | 0.000 |
| neonatal period-Intervention | 2.42 (1.461) | 0.38 (0.732) | 16.6 | 1.796 | 2.281 | 0.000 |
| neonatal period-Control | 1.95 (1.492) | 0.82 (1.152) | 11.15 | 0.93 | 1.328 | 0.000 |
| Components of bp-Intervention | 2.88 (1.415) | 1.03 (0.685) | 16.33 | 1.633 | 2.082 | 0.000 |
| Components of bp-Control | 1.95 (1.575) | 1 (1.189) | 10.74 | 0.781 | 1.131 | 0.000 |
| Antenatal services-Intervention | 1.55 (0.74) | 1.77 (0.587) | -3.288 | -0.352 | -0.088 | 0.000 |
| Antenatal services-Control | 1.07 (0.92) | 1.06 (0.83) | 0.169 | -0.117 | 0.139 | 0.000 |
| Knowledge on BRCR-Intervention | 14.1(5.761) | 4.43 (2.748) | 20.72 | 8.744 | 10.59 | 0.000 |
| Knowledge on BRCR-Control | 9.98 (6.654) | 5.5 (5.207) | 10.95 | 3.674 | 5.282 | 0.000 |

**Table 8. The relationship between respondents' characteristics and change in knowledge among pregnant women.**

| Variables | Change in BPCR knowledge scores | | | 95.0% CI | | p-value |
|---|---|---|---|---|---|---|
| | B | Beta | t | Lower | Upper | |
| *Education status* | | | | | | |
| No education (Constant) | 0.845 | | 7.656 | 0.628 | 1.062 | 0.000 |
| Primary education | 0.044 | 0.014 | 0.312 | -0.231 | 0.318 | 0.755 |
| Secondary education | -0.417 | -0.06 | -1.341 | -1.027 | 0.194 | 0.181 |
| *Groups* | | | | | | |
| Control group (Constant) | 0.473 | | 6.243 | 0.324 | 0.621 | 0.000 |
| CBCT group | 1.132 | 0.347 | 8.634 | 0.874 | 1.389 | 0.000 |
| *Age of respondent* | | | | | | |
| (Constant) | 0.603 | | 2.422 | 0.114 | 1.093 | 0.023 |
| Age of the respondent in years | 0.009 | 0.044 | 1.025 | -0.008 | 0.026 | 0.116 |
| *Ethnic group* | | | | | | |
| Fipa ethnicity (Constant) | 0.879 | | 10.597 | 0.716 | 1.042 | 0.000 |
| Mambwe ethnicity | -0.379 | -0.100 | -2.277 | -0.706 | -0.052 | 0.023 |
| other ethnic groups | 0.279 | 0.069 | 1.573 | -0.069 | 0.627 | 0.116 |
| (Constant) | -0.201 | | -0.599 | -0.862 | 0.459 | 0.549 |
| Age at marriage in years | 0.052 | 0.135 | 3.188 | 0.020 | 0.085 | 0.002 |
| *Characteristic of nearby health facility* | | | | | | |
| Health center (Constant) | 0.653 | | 4.138 | 0.343 | 0.962 | 0.000 |
| Dispensary | 0.239 | 0.059 | 1.376 | -0.102 | 0.58 | 0.169 |
| *Distance to a health facility* | | | | | | |
| Less than 1 kilometer (Constant) | 0.800 | | 8.375 | 0.612 | 0.988 | 0.000 |
| One to five kilometers | 0.104 | 0.034 | 0.748 | -0.170 | 0.378 | 0.455 |
| More than five kilometers | 0.057 | 0.011 | 0.252 | -0.389 | 0.503 | 0.801 |
| Ever heard about BPCR | | | | | | |
| No (Constant) | 1.204 | | 7.787 | 0.900 | 1.508 | 0.000 |
| Yes | -0.432 | -0.108 | -2.529 | -0.767 | -0.096 | 0.012 |

(β = 1.058, p<0.001) and the walking distance [more than five kilometers (β = -0.55, p< 0.05)] Table 10.

## Discussion

Birth preparedness in the developed world is a norm and almost all births are attended by skilled attendants. Ninety-nine percent of deliveries in Europe were assisted by skilled birth attendants in 2005 [29]. Similarly, in the same year, 98.9% of births were assisted by skilled birth attendants in North America and 85.5% in Oceania. In contrast, the proportion of skilled birth assistance in Asia was 59.1% and only 43.7% of women accessed skilled birth assistance in Africa [30].

Birth preparedness in developing countries, especially in South Asia and Sub-Saharan Africa, is unacceptably low [11–13] which enlightens the large proportion of up to 180 million women failing to access skilled birth assistance in these regions [31]. The majority of births which failed to access skilled birth assistance were from rural areas [31]. Low levels of knowledge about birth preparedness and complication readiness among pregnant women are among the factors which hinder birth preparedness and complication readiness in developing countries such as Tanzania.

The CBCT project was designed to address the challenge of low knowledge about birth preparedness in rural settings of Tanzania. In contrary to the existing system of antenatal sessions

**Table 9. The relationship between respondents' characteristics and change in knowledge among male partners.**

|  | Change in knowledge scores | | | 95.0% CI | | p- |
| --- | --- | --- | --- | --- | --- | --- |
| Variables | B | Beta | t | Lower | Upper | value |
| *No formal education (Constant)* | 0.958 |  | 7.911 | 0.72 | 1.196 | 0.000 |
| Primary education | -0.117 | -0.034 | -0.769 | -0.417 | 0.182 | 0.442 |
| Secondary education | -0.181 | -0.023 | -0.523 | -0.858 | 0.497 | 0.601 |
| (Constant) | 0.462 |  | 5.607 | 0.300 | 0.623 | 0.000 |
| CBCT group | 1.253 | 0.353 | 8.787 | 0.973 | 1.533 | 0.000 |
| (Constant) | 1.517 |  | 5.619 | 0.987 | 2.047 | 0.000 |
| Age of the respondent in years | -0.023 | -0.104 | -2.450 | -0.041 | -0.005 | 0.015 |
| (Constant) | 1.103 |  | 12.54 | 0.930 | 1.276 | 0.000 |
| Mambwe ethnicity | -0.841 | -0.212 | -4.925 | -1.177 | -0.506 | 0.000 |
| other ethnic groups | -0.23 | -0.046 | -1.075 | -0.650 | 0.190 | 0.283 |
| (Constant) | 0.671 |  | 1.744 | -0.085 | 1.428 | 0.082 |
| Age at marriage in years | 0.01 | 0.024 | 0.549 | -0.027 | 0.048 | 0.583 |
| (Constant) | 1.213 |  | 7.036 | 0.874 | 1.551 | 0.000 |
| Dispensary | -0.403 | -0.091 | -2.127 | -0.775 | -0.031 | 0.034 |
| (Constant) | 0.802 |  | 7.769 | 0.599 | 1.005 | 0.000 |
| One to five kilometers | 0.313 | 0.092 | 2.091 | 0.019 | 0.607 | 0.037 |
| More than five kilometers | -0.58 | -0.103 | -2.337 | -1.068 | -0.092 | 0.020 |
| (Constant) | 1.111 |  | 6.602 | 0.781 | 1.442 | 0.000 |
| Ever heard BP | -0.283 | -0.065 | -1.523 | -0.649 | 0.082 | 0.128 |

where expecting couples are expected to visit a health facility, the CBCT project was a community-based. Each expecting couples were trained at their home which created a conducive environment for learning.

**Table 10. Predictors of change in knowledge score among study respondents.**

|  | Predictor Variables | Change in knowledge score | | | 95.0% CI | | p-value |
| --- | --- | --- | --- | --- | --- | --- | --- |
|  |  | B | Beta | t | Lower | Upper |  |
| Pregnant | (Constant) | 0.356 |  | 1.431 | -0.133 | 0.844 | 0.153 |
| Women | CBCT group | 1.129 | 0.346 | 8.527 | 0.869 | 1.389 | 0.000 |
|  | Primary education | 0.019 | 0.006 | 0.143 | -0.244 | 0.283 | 0.887 |
|  | Secondary education | -0.512 | -0.07 | -1.773 | -1.078 | 0.055 | 0.077 |
|  | Mambwe ethnicity | -0.524 | -0.14 | -3.198 | -0.846 | -0.2 | 0.001 |
|  | Other ethnic groups | 0.257 | 0.064 | 1.5 | -0.08 | 0.595 | 0.134 |
|  | Nearby health facility dispensary | 0.286 | 0.071 | 1.647 | -0.055 | 0.628 | 0.100 |
|  | One to five kilometers | 0.25 | 0.08 | 1.892 | -0.01 | 0.51 | 0.059 |
|  | More than five kilometers | 0.309 | 0.061 | 1.348 | -0.141 | 0.759 | 0.178 |
|  | Ever heard dummy | -0.217 | -0.05 | -1.34 | -0.536 | 0.101 | 0.181 |
| Male | (Constant) | 1.197 |  | 3.368 | 0.499 | 1.895 | 0.001 |
| Partners | CBCT group | 1.058 | 0.298 | 6.668 | 0.746 | 1.369 | 0.000 |
|  | Age of the respondent in years | -0.013 | -0.06 | -1.51 | -0.031 | 0.004 | 0.132 |
|  | Mambwe ethnicity | -0.254 | -0.06 | -1.424 | -0.604 | 0.096 | 0.155 |
|  | Other ethnic groups | 0.023 | 0.005 | 0.109 | -0.391 | 0.437 | 0.914 |
|  | Nearby health facility dispensary | -0.29 | -0.07 | -1.486 | -0.674 | 0.093 | 0.138 |
|  | One to five kilometers | 0.125 | 0.037 | 0.865 | -0.159 | 0.409 | 0.387 |
|  | More than five kilometers | -0.55 | -0.1 | -2.17 | -1.048 | -0.05 | 0.03 |

Findings from this quasi experimental study design have shown that, the project significantly increased the knowledge mean scores on birth preparedness and complication readiness among both pregnant women and their male partners.

The magnitudes of increase in mean scores were large in both groups. This suggests that empowerment of local village workers is foundational towards improving knowledge on birth preparedness and complication readiness at the community level. This finding has been demonstrated in other settings.

In Nepal Mullany et al., [21] a group of pregnant women who had antenatal sessions with their husband showed to have increased knowledge on birth preparedness when compared to those who had antenatal session alone. Similar findings were also reported in other studies [17, 32].

The study also has found out that, despite the similarity in socio-demographic characteristics among male respondents in the intervention group and control group, there was significant increases in knowledge mean score among male respondents in the control group, as compared to male respondents in the intervention group at the pre-test assessment.

Similar findings were reported in similar interventional study done in Eritrea [17]. The significant difference at the pre-test assessment could have affected the magnitude of the effectiveness of the project on improving knowledge about birth preparedness and complication readiness among male respondents.

This is because despite of the difference between them, male partners in the control group having a significant higher mean score at the pre-test but still the comparison at the post-test has revealed a significant increase in knowledge among male partners in the intervention group.

Although there was a significant difference at the post-test assessment among both pregnant women and their male partners between the intervention and control groups, the pre-post assessment among both groups showed a significant mean score increase in knowledge mean scores.

The increase in knowledge in control group could be due to knowledge gained by the pregnant women and their male partners between the pre-test assessment and the post-test assessment. The pre-test assessment was performed prior to 24 weeks gestational age and the post-test assessment was completed after 42 days following childbirth.

The effect size was calculated to determine the strength of the difference which revealed a large effect size in both pregnant women and male partners, in the intervention group as compared to the medium effect size in the control group among both pregnant women and their male partners.

The study went further and determined the predictors of change in knowledge mean scores between the two surveys. The change scores were the new variable computed as the score of an individual at the post-test assessment minus the score at the pre-test assessment.

The predictors of change among pregnant women were the intervention and pregnant women's ethnic group. The pregnant women's ethnic group could have influenced the change because most of pregnant women in the intervention group, were more of Fipa ethnicity than Mambwe, which draws a conclusion that the change in knowledge score among pregnant women were due to intervention. The predictors of change among male respondents were the intervention and walking distance to nearby health facility. Change in knowledge increased as the distance to a nearby health facility decreases.

The possible reason for this could be people learn better when they have previous existing memory of newly introduced subject matter [33]. Studies have revealed that, walking distance to a nearby health facility is a barrier to male involvement in maternal services utilization [34] which this intervention study revealed. Men who reside far were less likely to be knowledgeable about birth preparedness and complication readiness.

The intervention significantly increases the knowledge about birth preparedness mean scores among both pregnant women and their male partners. Although this interventional study established causal effect relationship but still it had some limitations. One of the limitations was lack of random assignment of couples. This could have limited the comparability of the two groups (intervention group and control group). The limitation was minimized by use of probability sampling technique in selection of couples in both groups. Furthermore, the limitation was minimized by matching pregnant women in both groups using age in five years groups and parity. Authors decided to use these two criteria to avoid having one group with younger pregnant women and the other group with older pregnant women.

Another limitation of this study was the recall bias. After training couples were expect to recall were expected to recall and mention the danger signs during pregnancy, labor, post delivery and neonatal; they were also expected to recall what to prepare for childbirth. The limitation was minimized by having a control group and supplementation of the sessions with pictorial reference materials.

## Conclusion

This interventional study focused on knowledge empowerment and behavior change among expecting couples was effective in improving knowledge about birth preparedness among both pregnant women and their male partners. Knowledge retention at post-test assessment showed a remarkable difference. This indicates that, the project had an impact on improving core knowledge on birth preparedness and complication readiness.

## Supporting information

**S1 File.**
(SAV)

**S2 File.**
(DOC)

**S3 File.**
(DOC)

## Acknowledgments

The authors thank the University of Dodoma for providing ethical clearance for this study. We also thank the Rukwa Region administration for allowing us to conduct the study, also we thank the study respondents for volunteering to participate in this study.

## Author Contributions

**Conceptualization:** Fabiola V. Moshi.

**Data curation:** Fabiola V. Moshi.

**Formal analysis:** Fabiola V. Moshi.

**Investigation:** Fabiola V. Moshi.

**Methodology:** Fabiola V. Moshi.

**Project administration:** Fabiola V. Moshi.

**Supervision:** Stephen M. Kibusi, Flora Masumbuo Fabian.

**Writing – original draft:** Fabiola V. Moshi.

**Writing – review & editing:** Fabiola V. Moshi.

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
