## [Decision Letter · Decision Letter 0]

26 Aug 2020

PONE-D-19-30480

The Impact of Community Based Continuous Training Project on Improving Couples’ Knowledge on Birth Preparedness and Complication Readiness in Rural Setting Tanzania; A Controlled Quasi-Experimental Study

PLOS ONE

Dear Dr. Moshi,

Thank you for submitting your manuscript to PLOS ONE. After careful consideration, we feel that it has merit but does not fully meet PLOS ONE’s publication criteria as it currently stands. Therefore, we invite you to submit a revised version of the manuscript that addresses the points raised during the review process.

We look forward to receiving your revised manuscript.

Kind regards,

Florian Fischer

Academic Editor

PLOS ONE

Journal Requirements:

3.Thank you for stating the following financial disclosure:

 [NO - Include this sentence at the end of your statement: The funders had no role in study design, data collection and analysis, decision to publish, or preparation of the manuscript.].

4. Please include a caption for figure 4 (Figure 3 caption included 2x).

5. We note you have included a table to which you do not refer in the text of your manuscript. Please ensure that you refer to Table 2 in your text; if accepted, production will need this reference to link the reader to the Table.

Reviewers' comments:

Reviewer's Responses to Questions

**Comments to the Author**

1. Is the manuscript technically sound, and do the data support the conclusions?

Reviewer #1: Yes

Reviewer #2: Yes

2. Has the statistical analysis been performed appropriately and rigorously? 

Reviewer #1: Yes

Reviewer #2: Yes

3. Have the authors made all data underlying the findings in their manuscript fully available?

Reviewer #1: Yes

Reviewer #2: Yes

4. Is the manuscript presented in an intelligible fashion and written in standard English?

Reviewer #1: Yes

Reviewer #2: Yes

5. Review Comments to the Author

Reviewer #1: Introduction:

Line 53 and 54: should be clear and specific where these deaths occurred

Study design:

Well done

Setting:

Well done

Sampling technique:

Line 168: correct the name from Sumawanga to Sumbawanga

Line 171: what was the interval used in the systematic sampling in the selection of the households?

Line 172: state the inclusion criteria for the study population

Line 173: who was responsible for the assessment of signs and symptoms of the pregnancy in each household visited? This needs to be clarified and the signs and the symptoms assessed as well.

Line 185: check the value of the power at 90% is not 0.84, please confirm and correct.

Line 203: the couples were only visited for two days irrespective of the gestation age, how will this intervention have impact for a couple visited at early stages of gestation age?

Line 204: you skipped what happened in step four and went to fifth step, please correct

Data collection procedure

Line 210: was there any consenting process conducted to the participants? This needs to be stated before the start of data collection. Was the data collected using paper questionnaire or electronically? The author should clarify this. Were there also participants below age 18?

Data Processing and analysis

the results were well described, how were the assumptions of linear regression verified. the authors also needs to explain whether any statistical method was used to select variables for the multivariable analysis

Results:

Table 1: what was the lower age limit of the participants that were interviewed?

Table 1: Please indicate the exactly P-values whether significant of or not instead of leaving them blank and showing significant one with asterisk. Indicate the number of participants in each of the groups, and the n (%) at the top of each column to show what the values in the table mean

Table 2: add the p-values, show the lower limit of the gestation age

Table 3: add the exact p-values to the table

Discussion:

Well explained

Were there any strength and limitation for this study? This needs to be explained.

Reviewer #2: Thank you for the opportunity to review the manuscript entitled, "The Impact of Community Based Continuous Training Project on Improving Couples' Knowledge on Birth Preparedness and Complication Readiness in Rural Setting Tanzania: A controlled quasi-experimental study." The authors present the effect of the CBCT training on knowledge and complication readiness in rural Tanzania. Based on data presented, there appears to be improvement on knowledge scores for pregnant women and their male partners. The paper and intervention are interesting; however, there are some key questions that need to be addressed.

MAJOR

1. The authors published a protocol paper with the following citation: "Moshi FV, Kibusi SM, Fabian F. The Effectiveness of Community-Based Continuous Training on Promoting Positive Behaviors towards Birth Preparedness, Male Involvement, and Maternal Services Utilization among Expecting Couples in Rukwa, Tanzania: A Theory of Planned Behavior Quasi-Experimental Study. J Environ Public Health. 2018;2018:1293760. Published 2018 Sep 27. doi:10.1155/2018/1293760" However, the authors do not reference the protocol paper in this manuscript. It is unclear if these are full related studies. If the paper previously published, it would be helpful to reference the methods and then reduce the detail in the introduction and methods sections. For example, the sample size calculation and equations could be removed and referenced.

2. Please clarify how couples were randomized to receive the intervention or not; or were villages randomized to the intervention? It is clear from the description how the districts and villages were selected.

3. Please add details in the Methods about the timing of data collection with respect to the pre-test assessment and post-test assessment. These details are included in the discussion (Lines 403-405). Can the authors discuss potential recall bias or influence of the birth experience and outcomes on the post-test results?

4. Given that the authors interviewed the same couples pre and post intervention, it seems a difference-in-differences analysis would be appropriate. Can the authors discuss why they have not used this approach? Further it would streamline the Tables 3-7. Figures 3 and 4 also present similar information to Tables 3-7; please consider putting one or the other in the supplementary information.

MINOR:

1. In the abstract, please consider adding the baseline knowledge score so readers can understand more clearly the impact of the intervention.

2. In line 96, there is a reference "(martin)" that needs a full citation. Please update.

3. In lines 104 to 121, the authors present several programs for improving pregnant women's outcomes. This contrast and comparison presented may be more useful in the Discussion section to set the results from this research in context.

4. In lines 196-197, please provide clarification as how the doubling of the control arm increases comparability.

5. Please provide more clarity to the actual intervention that was applied; lines 198-207 are more about the measurement of the intervention, but not about the intervention for the couple. This information would be helpful to the readers.

6. Table 4: Please clarify if the mean (SD) knowledge is a score or mean number of danger signs assessed. Further, is there a range associated with each assessment? Further, please limit to 2 significant digits for the mean and SD of scores.

7. Figures 3 and 4 presentation as line charts make the reader compare scores across domain as opposed to pre/post scores. Consider a bar chart for pre/post scores next to each other.

6. PLOS authors have the option to publish the peer review history of their article (what does this mean?). If published, this will include your full peer review and any attached files.

Reviewer #1: **Yes: **James Orwa

Reviewer #2: No

---

## [Author Response · Author response to Decision Letter 0]

23 Oct 2020

Response to Reviewers’ and Editor’s comments

Editor’s Comments

Comment 1

Please ensure that your manuscript meets PLOS ONE's style requirements, including those for file naming. The PLOS ONE style

Response 1

The manuscript meets PLOSE ONE style

Comment 2

Please include additional information regarding the survey or questionnaire used in the study and ensure that you have provided sufficient details that others could replicate the analyses. For instance, if you developed a questionnaire as part of this study and it is not under a copyright more restrictive than CC-BY, please include a copy, in both the original language and English, as Supporting Information.

Response 2

The questionnaire was adopted and modified from Jhpiego. Monitoring birth preparedness and complication readiness: tools and indicators for maternal and newborn health. 2004;1–338. Available from: http://pdf.usaid.gov/pdf_docs/PNADA619.pdf . The tool in both language (English and Swahili) is attached as supporting Information

Comment 3

[NO - Include this sentence at the end of your statement: The funders had no role in study design, data collection and analysis, decision to publish, or preparation of the manuscript.]. At this time, please address the following queries:

 Please clarify the sources of funding (financial or material support) for your study. List the grants or organizations that supported your study, including funding received from your institution.

 State what role the funders took in the study. If the funders had no role in your study, please state: “The funders had no role in study design, data collection and analysis, decision to publish, or preparation of the manuscript.”

 If any authors received a salary from any of your funders, please state which authors and which funders.

 If you did not receive any funding for this study, please state: “The authors received no specific funding for this work.” Please include your amended statements within your cover letter; we will change the online submission form on your behalf.

Response

This work is part of my PhD project. The University of Dodoma sponsored the tuition fee and field allowance. Publication fee was not covered in the sponsorship. 

The funders had no role in decision to publish, or preparation of the manuscript

The authors received no specific funding for this work

Comment 4

Please include a caption for figure 4 (Figure 3 caption included 2x).

Response 4

The caption for figure 4 is included; the repeated figure three was a typing error. It was meant to be figure 4

Comment 5

We note you have included a table to which you do not refer in the text of your manuscript. Please ensure that you refer to Table 2 in your text; if accepted, production will need this reference to link the reader to the Table.

Response 5

Table 2 is referred in the text line 260

Comments from reviewer 1

Comment 1

Line 53 and 54: should be clear and specific where these deaths occurred

Response 1

Authors agree with the comment and correction is made

Comment 2

Line 168: correct the name from Sumawanga to Sumbawanga

Response 2

Authors agree with the correction and it is done

Comment 3

Line 171: what was the interval used in the systematic sampling in the selection of the households?

Response 3

No specific interval was used but rather pre determine direction of selection of household after random selection of the first household

Comment 4

Line 172: state the inclusion criteria for the study population

Response 4

The inclusion criteria are mentioned

Comment 5

Line 173: who was responsible for the assessment of signs and symptoms of the pregnancy in each household visited? This needs to be clarified and the signs and the symptoms assessed as well.

Response 5

Research assistants who were nurses did the interview and confirmed the test

Comment 6

Line 185: check the value of the power at 90% is not 0.84, please confirm and correct.

Response 6

The sample size calculation is removed as the portion is published

Comment 7

Line 203: the couples were only visited for two days irrespective of the gestation age; how will this intervention have impact for a couple visited at early stages of gestation age?

Response 7

The study reports a part of the study

One of the components of the intervention was followed up. They were followed and assessed for three main behaviors which were birth preparedness, male involvement and use of health facilities for antenatal services, delivery services and postnatal services

In addition they were supplied with pictorial reference materials

Comment 8

Line 204: you skipped what happened in step four and went to fifth step, please correct 

Response 8

What happened in step four is included

Comment 9

Line 210: was there any consenting process conducted to the participants? This needs to be stated before the start of data collection. Was the data collected using paper questionnaire or electronically? The author should clarify this. Were there also participants below age 18? Before participation into the study couples were counseled and upon their consent, they were included in the study (Line 244-Line 246)

Comment 9

Upon consent to participate, there record were taken to the nearby health facility where their behavior were recorded as they acted

Response 9

No electronic data collection was used but an interviewer administered questionnaire 

Comment 10

The results were well described, how were the assumptions of linear regression verified. the authors also needs to explain whether any statistical method was used to select variables for the multivariable analysis 

Response 10

Variables which showed significant correlation with knowledge about birth preparedness were entered into the regression model

Comment 11

Table 1: what was the lower age limit of the participants that were interviewed?

Response 11

The lowest age was 16 years. 

Comment 12

Table 1: Please indicate the exactly P-values whether significant of or not instead of leaving them blank and showing significant one with asterisk. Indicate the number of participants in each of the groups, and the n (%) at the top of each column to show what the values in the table mean

Response 12

The column of p value is removed in this table because the aim of the table is to describe the characteristics of the study respondents.

Authors agree with the observation and n(%) are added in each column

Comment 13

Table 2: add the p-values, show the lower limit of the gestation age

Response 13

The p-values and lower gestation age is added to the table

Comment 14

Table 3: add the exact p-values to the table 

Response 14

The exact p-values is added to the table

Comment 15

Were there any strength and limitation for this study? This needs to be explained.

Response 15

Strength and limitation of the study is added in the last two paragraphs after discussion (Line 236- Line 449)

Comments from reviewer 2 

Comment 1

The authors published a protocol paper with the following citation: "Moshi FV, Kibusi SM, Fabian F. The Effectiveness of Community-Based Continuous Training on Promoting Positive Behaviors towards Birth Preparedness, Male Involvement, and Maternal Services Utilization among Expecting Couples in Rukwa, Tanzania: A Theory of Planned Behavior Quasi-Experimental Study. J Environ Public Health. 2018;2018:1293760. Published 2018 Sep 27. doi:10.1155/2018/1293760" However, the authors do not reference the protocol paper in this manuscript. It is unclear if these are full related studies. If the paper previously published, it would be helpful to reference the methods and then reduce the detail in the introduction and methods sections. For example, the sample size calculation and equations could be removed and referenced. 

Response 1

Yes, the protocol of the study is published. This manuscript is a part of the findings from the protocol. Authors agree with the suggestion and the protocol is cited in the sample size calculation 

Comment 2

Clarify how couples were randomized to receive the intervention or not; or were villages randomized to the intervention? It is clear from the description how the districts and villages were selected.

Response 2

Randomization was not possible in this intervention; there were a lot of limitations one of which was contamination of the intervention. Applying random assignment to neighbors was not possible

It was a quasi experimental study where only the districts were randomly assigned to either intervention or control 

Comment 3

Please add details in the Methods about the timing of data collection with respect to the pre-test assessment and post-test assessment. These details are included in the discussion (Lines 403-405). Can the authors discuss potential recall bias or influence of the birth experience and outcomes on the post-test results?

Response 3

A sentence is added in the data collection to explain the timing for data collection for both pre test assessments (Line 223 to Line 225) 

Limitation of the study is added after the discussion where recall bias as one of the limitation is discussed (Line 436 to Line 448)

Comment 4

Given that the authors interviewed the same couples pre and post intervention, it seems a difference-in-differences analysis would be appropriate. Can the authors discuss why they have not used this approach? Further it would streamline the Tables 3-7. Figures 3 and 4 also present similar information to Tables 3-7; please consider putting one or the other in the supplementary information.

Response 4

Yes the same couple was assessed pre pre-post in both groups. Difference-in-difference analysis was used.

The result section is organized into three main parts. The first part is the comparison between groups pre- assessment and post assessment. The second part of the results is the comparison within the group, pre- post assessment differenced between both groups. The third part is the analysis of the predictors of difference in mean score. For every respondent the difference score was computed. This score was treated as a new variable. Predictors of change were established using multiple liner regression.

Authors think allowing all tables to be in the study could still hold water but we are looking forward for more opinion

Comment 5

In the abstract, please consider adding the baseline knowledge score so readers can understand more clearly the impact of the intervention.

Response 5

Authors agree with the suggestion

Comment 6

In line 96, there is a reference "(martin)" that needs a full citation. Please update.

Response 6

The reference is inserted

Comment 7

In lines 104 to 121, the authors present several programs for improving pregnant women's outcomes. This contrast and comparison presented may be more useful in the Discussion section to set the results from this research in context.

Response 7

The introduction of several interventions in the background was placed there to introduce the current intervention. If moved can create discontinuity in the background

Comment 8

In lines 196-197, please provide clarification as how the doubling of the control arm increases comparability.

Response 8

Authors aimed at having groups with similar characteristics; avoiding having one wing with older population and other group with younger population

Comment 9

Please provide more clarity to the actual intervention that was applied; lines 198-207 are more about the measurement of the intervention, but not about the intervention for the couple. This information would be helpful to the readers.

Response 9

Actual clarification is added line 218-line 223

Comment 10

Table 4: Please clarify if the mean (SD) knowledge is a score or mean number of danger signs assessed. Further, is there a range associated with each assessment? Further, please limit to 2 significant digits for the mean and SD of scores.

Response 10

It is a mean knowledge score. One point for each recalled danger sign or what to prepare for birth/emergency

Decimal points are restricted to two places

Comment 11

Figures 3 and 4 presentation as line charts make the reader compare scores across domain as opposed to pre/post scores Consider a bar chart for pre/post scores next to each other

Response 11

Authors agree with the suggestion and the figure is changed to chart for pre/post

---

## [Decision Letter · Decision Letter 1]

17 Nov 2020

PONE-D-19-30480R1

The Impact of Community Based Continuous Training Project on Improving Couples’ Knowledge on Birth Preparedness and Complication Readiness in Rural Setting Tanzania; A Controlled Quasi-Experimental Study

PLOS ONE

Dear Dr. Moshi,

Thank you for submitting your manuscript to PLOS ONE. After careful consideration, we feel that it has merit but does not fully meet PLOS ONE’s publication criteria as it currently stands. Therefore, we invite you to submit a revised version of the manuscript that addresses the points raised during the review process.

Please find below some minor comments raised by Reviewer 1. We ask you to incorporate these suggestions, before we can accept the manuscript.

We look forward to receiving your revised manuscript.

Kind regards,

Florian Fischer

Academic Editor

PLOS ONE

Reviewers' comments:

Reviewer's Responses to Questions

**Comments to the Author**

1. If the authors have adequately addressed your comments raised in a previous round of review and you feel that this manuscript is now acceptable for publication, you may indicate that here to bypass the “Comments to the Author” section, enter your conflict of interest statement in the “Confidential to Editor” section, and submit your "Accept" recommendation.

Reviewer #1: (No Response)

Reviewer #2: All comments have been addressed

2. Is the manuscript technically sound, and do the data support the conclusions?

Reviewer #1: Partly

Reviewer #2: Yes

3. Has the statistical analysis been performed appropriately and rigorously? 

Reviewer #1: Yes

Reviewer #2: Yes

4. Have the authors made all data underlying the findings in their manuscript fully available?

Reviewer #1: Yes

Reviewer #2: Yes

5. Is the manuscript presented in an intelligible fashion and written in standard English?

Reviewer #1: Yes

Reviewer #2: Yes

6. Review Comments to the Author

Reviewer #1: Most of the previous comments are addressed, just a few clarity needed

The author mentioned that the first household was selected using a random sampling approach,they selected the subsequent households using systematic sampling but no interval stated. this still needs clarity.

The author have removed the sample size calculation but cited the published propotocol, it would have been informative if a brief on this is given.

verbal consent was done for women below 18 years, was there a justification why there was no assent? in some cases this group have been treated as mature minors and could provide their own consent. is there a justification why this is no the case in this population??

Reviewer #2: Thank you for addressing the comments and revising the manuscript. I also appreciate that the files have been added in supplemental information. Please note that during the proofing process, please conduct a thorough review, there are some errors that won't be caught by spell checker (i.e. line 240, Match for March, etc)

7. PLOS authors have the option to publish the peer review history of their article (what does this mean?). If published, this will include your full peer review and any attached files.

Reviewer #1: **Yes: **James Orwa

Reviewer #2: No

---

## [Author Response · Author response to Decision Letter 1]

3 Dec 2020

Point to point response to reviewers’ comments

Reviewer 1

Comment 1

The author mentioned that the first household was selected using a random sampling approach, they selected the subsequent households using systematic sampling but no interval stated. this still needs clarity.

Response 1

Yes, the first household was picked randomly followed by systematic sampling. No interval was set but researchers move to already pre-determined direction. Each visited households were assessed for eligibility, if eligible was selected and next household was visited, if not eligible was skipped and next household was visited until the required sample size was reached 

Comment 2

The authors have removed the sample size calculation but cited the published protocol, it would have been informative if a brief on this is given.

Response 2

Authors agree with the recommendation and the sample size calculation is added line 168-189

Comment 3

Verbal consent was done for women below 18 years, was there a justification why there was no assent? in some cases this group have been treated as mature minors and could provide their own consent. is there a justification why this is not the case in this population??

Response 3

Authors agree with the comment, it is true they are also taken as mature even in this population. What was done in the project were to seek consent from pregnant women then from their male partners; It was until both of them agree to participate then they were included in the study. The sentence is removed.

Reviewer 2

Comment 1

Thank you for addressing the comments and revising the manuscript. I also appreciate that the files have been added in supplemental information. Please note that during the proofing process, please conduct a thorough review, there are some errors that won't be caught by spell checker (i.e. line 240, Match for March, etc.)

Response 1

Line 240 is corrected

---

## [Editor Report · Decision Letter 2]

18 Dec 2020

The Impact of Community Based Continuous Training Project on Improving Couples’ Knowledge on Birth Preparedness and Complication Readiness in Rural Setting Tanzania; A Controlled Quasi-Experimental Study

PONE-D-19-30480R2

Dear Dr. Moshi,

We’re pleased to inform you that your manuscript has been judged scientifically suitable for publication and will be formally accepted for publication once it meets all outstanding technical requirements.

Kind regards,

Florian Fischer

Academic Editor

PLOS ONE
---

## [Editor Report · Acceptance letter]

23 Dec 2020

PONE-D-19-30480R2 

The Impact of Community Based Continuous Training Project on Improving Couples’ Knowledge on Birth Preparedness and Complication Readiness in Rural Setting Tanzania; A Controlled Quasi-Experimental Study  

Dear Dr. Moshi:

I'm pleased to inform you that your manuscript has been deemed suitable for publication in PLOS ONE. Congratulations! Your manuscript is now with our production department. 

Kind regards, 

on behalf of

Dr. Florian Fischer 

Academic Editor

PLOS ONE